# Multiple Socioeconomic Circumstances and Initiation of Cardiovascular Medication among Ageing Employees

**DOI:** 10.3390/ijerph181910148

**Published:** 2021-09-27

**Authors:** Aino Salonsalmi, Elina Mauramo, Eero Lahelma, Ossi Rahkonen, Olli Pietiläinen

**Affiliations:** Department of Public Health, University of Helsinki, 00014 Helsinki, Finland; elina.mauramo@helsinki.fi (E.M.); eero.lahelma@helsinki.fi (E.L.); ossi.rahkonen@helsinki.fi (O.R.); olli.k.pietilainen@helsinki.fi (O.P.)

**Keywords:** socioeconomic factors, cardiovascular diseases, lipid medication, hypertension medication

## Abstract

There are persisting socioeconomic differences in cardiovascular diseases, but studies on socioeconomic differences in the initiation of cardiovascular medication are scarce. This study examined the associations between multiple socioeconomic circumstances and cardiovascular medication. The Helsinki Health Study baseline survey (2000–2002) of 40–60-year-old employees was linked with cardiovascular medication data from national registers. The analyses included 5805 employees concerning lipid medication and 4872 employees concerning hypertension medication. Medication purchases were followed for 10 years. The analyses were made using logistic regression, and the odds ratios (ORs) and their 95% confidence intervals (CIs) were calculated for childhood, conventional and material socioeconomic circumstances. Low parental education showed an association with lipid medication among women only (OR 1.34, 95% CI 1.11–1.61), whereas childhood economic difficulties showed more widespread associations. Low education and occupational class were associated with an increased risk of both hypertension (education: OR 1.58, 1.32–1.89; occupational class: 1.31, 1.08–1.59) and lipid medication (education: 1.34, 1.12–1.61; occupational class: 1.38, 1.13–1.67). Rented housing (1.35, 1.18–1.54 for hypertension medication; 1.21, 1.05–1.38 for lipid medication) and current economic difficulties (1.59, 1.28–1.98 for hypertension medication; 1.35, 1.07–1.71 for lipid medication) increased the risk. Several measures of socioeconomic circumstances acting at different stages of the life course were associated with cardiovascular medication, with individuals in disadvantageous socioeconomic circumstances having elevated risks.

## 1. Introduction

Cardiovascular mortality has fallen substantially during the recent decades in Finland, but cardiovascular diseases are still the second-most common cause of death [1] and the third-most common cause for disability retirement [2]. Additionally, globally, cardiovascular mortality has declined in wealthy countries [3], but cardiovascular diseases remain a major cause of premature death and disability in all regions of the world [4]. Socioeconomic differences in cardiovascular diseases are well-known and have persisted or even widened in Finland, as well as in other countries where cardiovascular mortality has declined [3,5].

The risk factors for cardiovascular diseases include hypertension and unfavourable blood lipid levels. The risk of cardiovascular events such as stroke and myocardial infarction can be reduced by hypertension and lipid medication. The most commonly used lipid medication—namely, statin therapy and treatment with any commonly used regimen for hypertension—reduce the risk of major cardiovascular events [6,7]. It has been suggested that preventions targeting risk reductions in high-risk individuals might widen the socioeconomic health differences if individuals in more advantageous socioeconomic circumstances are more likely to be on medication [8]. Studies of individuals with already diagnosed cardiovascular disease and adherence to medication have often reported either no associations or increasing medication or adherence with advantageous socioeconomic circumstances [9,10,11], thus partly supporting the assumption.

Studies on the initiation of cardiovascular medication are scarce and with inconclusive results. A Norwegian study found that the highest educated had the lowest probability of statin treatment [12]. The educational differences disappeared after adjustment for known cardiovascular risk factors at the baseline, suggesting that statins were prescribed according to cardiovascular risk independent of education. Danish and Finnish studies of individuals with diabetes found that a lower income was associated with a lower probability of statin prescribing [13,14]. A Norwegian cross-sectional study reported that, among presumed healthy women, statin medication was less common among the highly educated, whereas, among men, there were no differences [15]. A UK retrospective study found a trend towards an increased prescribing of lipid medication to patients from more deprived areas in both eligible and ineligible patients [16]. Studies of hypertension medication are particularly rare, but an Italian study found high income to be associated with decreased medication [17]. The medication was also decreased among people with low incomes compared to people with mid-level incomes [17].

Socioeconomic health differences can be regarded as a product of the unequal distribution of both material and behavioural health-endangering exposures and health-protective resources between socioeconomic groups acting over the course of life [18,19]. Socioeconomic circumstances cannot thus be fully captured by a single measure, but instead, a multi-domain approach is needed. Different measures of socioeconomic circumstances portray ranking in society and different subdomains of the umbrella concept of socioeconomic circumstances. Single measures of socioeconomic circumstances portray different exposures and resources, which, in turn, influence health through lifestyles, health behaviours, work and living conditions, and seeking and access to care [20]. Thus, divergent etiological processes likely lie behind the association of each socioeconomic measure and health. Although cardiovascular diseases develop over the entire course of life, previous studies have focused on conventional measures of socioeconomic circumstances—namely, education, occupational class, and income. We extend the socioeconomic circumstances to the life course and further material factors.

This study aimed to examine whether socioeconomic circumstances are associated with the initiation of lipid and hypertension medication among ageing employees. We used a broad framework of socioeconomic circumstances, including childhood socioeconomic circumstances, conventional measures of socioeconomic circumstances, and material circumstances.

## 2. Materials and Methods

### 2.1. Data

This study is part of the Helsinki Health Study of employees of the city of Helsinki in Finland [21]. The city of Helsinki is the largest employer in Finland, with about 37,000 employees engaging in various jobs, such as teachers, lawyers, doctors, nurses, cleaners, bus drivers, and garden workers [22]. The majority of the employees (76%) are women, as is the case elsewhere in the Finnish municipal sector.

The baseline survey was conducted by postal questionnaires in 2000, 2001, and 2002 among employees who reached the ages of 40, 45, 50, 55, or 60 during each year. The questionnaire consisted of questions, for example, on sociodemographic and socioeconomic factors [23], health, health behaviours, and working conditions. The questionnaire was sent to 13,346 employees—of whom, 8960 participated yielded a response rate of 67%. The response rate tended to be higher for employees who were older, in higher occupational classes, and with less sickness absences during the study year. The differences were, however, minor and not fully consistent [21,24].

The data on prescribed reimbursed medication purchases were derived from the registers of the Social Insurance Institution of Finland and the data on hospitalisations from the registers of the Finnish National Institute for Health and Welfare. The data on prescribed medication included the date of purchase and the type of prescribed medication. The questionnaire data were linked with the register data among respondents who consented to the data linkage (*n* = 6603, 74% of the participants). The follow-up started from the day of returning the baseline questionnaire and continued for 10 years. Three hundred and fifty-three participants had purchased lipid and 1365 hypertension medication during the three years preceding the baseline survey, and as the aim was to examine new cases, these participants were excluded from the corresponding analyses. After exclusions due to prior medication and missing data on covariates, this study included 5805 employees (1233 men and 4572 women) for the analyses on lipid medication and 4872 employees (1052 men and 3820 women) on hypertension medication. Due to item nonresponses on measures of the socioeconomic circumstances, the final analyses included slightly less participants.

The Ethics Committee of the Department of Public Health at the University of Helsinki and the health authorities of the city of Helsinki approved the study.

### 2.2. Socioeconomic Circumstances

Parental education was divided into three classes: ‘low’ (elementary school or part of it), ‘mid-level’ (intermediate or vocational school), and ‘high’ (matriculation examination or college-level training or polytechnic or university degree). Parental education was based on either maternal or paternal education, whichever was higher. Childhood economic difficulties were inquired with a single-item question asking if there were serious economic difficulties in the family before the respondent turned 16 years. The respondents’ own education was divided into three classes: ‘low’ (elementary school or intermediate school); ‘mid-level’ (vocational school, matriculation examination, or college-level training); and ‘high’ (polytechnic or university degree). The occupational class was divided into four classes according to the job title: managers and professionals such as teachers and physicians, semi-professionals such as nurses and foremen, nonmanual employees such as clerical employees and child minders, and manual workers such as cleaning workers. The monthly household income was divided into four categories. Housing tenure was dichotomised into owner occupiers and renters. Current economic difficulties were asked by questions of difficulties in paying bills and having enough money to buy food or clothing to one’s family, and a combined variable with three categories was formed: no, occasional, and frequent difficulties. (Table 1).

### 2.3. Cardiovascular Medication

Purchases of prescribed, reimbursed medication affecting the cardiovascular system were classified according to the Anatomical Therapeutic Chemical (ATC) system by the World Health Organization [25]. Cardiovascular medication was divided into lipid-modifying agents (ATC-code C10) and medication having hypertension as the indication: antihypertensives (C02), diuretics (C03), beta-blocking agents (C07), calcium-channel blockers (C08), and agents acting on the renin–angiotensin system (C09).

### 2.4. Covariates

The sociodemographic covariates consisted of age and gender. The body mass index (BMI) was calculated from self-reported data on height and weight and divided into three groups: under 25, between 25 and 30, and above 30 kg/m^2^. Leisure time physical activity was measured by four questions from which metabolic equivalent values (MET) were calculated. Smoking was classified as current smoking and non-smoking. Alcohol problems were measured by the CAGE scale (cutting down on alcohol, annoyed by criticism, feeling guilty, and in need of an eye-opener) [26]. Marital status was divided into three categories: single, married or cohabiting, and divorced or widowed. Mental and physical strenuousness of the work were both inquired by single-item questions. Mental health was measured by the emotional well-being scale of the RAND 36-item health survey [27]. The emotional well-being subscale was divided into quartiles. Hospitalisation due to cardiovascular disease was derived from the national register data. There were 317 hospitalisations in the dataset concerning hypertension medication and 411 concerning lipid medication.

### 2.5. Statistical Methods

The associations between socioeconomic circumstances and hypertension and lipid medication were analysed by a logistic regression analysis. The odds ratios (ORs) and their 95% confidence intervals (95% CIs) for having at least one reimbursed medication purchase during a 10-year follow-up were calculated. First, a base model adjusted for age and gender was fitted for each variable of socioeconomic circumstances. Then, other covariates were added one by one to the base models: first, the body mass index and health behaviours; next, the marital status, working conditions, and mental health; and finally, hospitalisation due to cardiovascular diseases. The interactions for gender and medication purchases were tested, and statistically significant interactions were found only for childhood economic difficulties and hypertension medication (*p* = 0.0287) and for parental education and lipid medication (*p* = 0.0422). Thus, these models are presented separately for men and women, and otherwise, the genders are pooled. SAS statistical program version 9.4 software (SAS Institute, Chicago, IL, USA) was used to perform the analyses.

## 3. Results

Half of the participants’ parental education was low, and nearly one-fifth reported having experienced childhood economic difficulties (Table 1). Half of the participants had mid-level education, whereas managers and professionals and nonmanual employees were the most common occupational classes. Two-thirds were owner occupiers, and 8% had economic difficulties frequently. Thirty-three percent of the participants received hypertension medication and 24% lipid medication during the 10-year follow-up. The prevalence of cardiovascular medication varied by socioeconomic circumstances, with individuals in more advantageous circumstances having a lower prevalence.

Childhood economic difficulties were associated with an increased risk of hypertension medication among men only (OR 1.81, 95% CI 1.29–2.55) (Table 2). The analyses showed associations with hypertension medication both for mid-level (1.21, 1.05–1.40) and low (1.58, 1.32–1.89) education in comparison to high education. Additionally, occupational class was associated with an increased risk of hypertension medication (1.25, 1.07–1.46 for nonmanual employees and 1.31, 1.08–1.59 for manual workers), whereas, for household income, a slightly elevated risk was found only in the second-lowest income group (1.21, 1.02–1.43). Living in a rented housing (1.35, 1.18–1.54) and occasional (1.19, 1.04–1.35) or frequent (1.59, 1.28–1.98) economic difficulties were associated with a higher risk for hypertension medication. Adjustment for the body mass index and health behaviours abolished part of the associations and attenuated the rest of them, whereas adjusting for marital status, working conditions, and general mental health and for hospitalisation due to cardiovascular disease had only minimal contributions.

Low parental education was associated with an increased risk of lipid medication among women only (1.34, 1.11–1.61) (Table 3). Childhood economic difficulties were associated with lipid medication (1.26, 1.08–1.49). There were associations with lipid medication both for own education (1.34, 1.12–1.61 for low education) and occupational class (1.38, 1.13–1.67 for manual workers). Concerning household income, only individuals in the lowest income group had an increased risk (1.22, 1.01–1.46). Living in a rented housing (1.21, 1.05–1.38) and economic difficulties (1.35, 1.07–1.71 for frequent economic difficulties) were associated with lipid medication. Adjustment for the body mass index and health behaviours mainly abolished the associations. Adjustment for marital status, working conditions, and mental health and further adjustment for hospitalisation due to cardiovascular disease had only minor contributions.

## 4. Discussion

This study examined the associations of multiple socioeconomic circumstances with the initiation of cardiovascular medication. No single measure of socioeconomic circumstances was paramount. Disadvantaged childhood and conventional and material socioeconomic circumstances were all associated with cardiovascular medication, with employees in disadvantageous socioeconomic circumstances having higher risk. A gradient was observed for own education and for current economic difficulties and, to a lesser extent, for occupational class.

There is a large body of literature reporting poorer cardiovascular health among people in disadvantageous socioeconomic circumstances. Previous studies have suggested that the prevention and treatment of cardiovascular diseases might add to socioeconomic differences. For example, coronary revascularisation [28,29] and guideline-recommended acute care of stroke [30] are more common among patients in advantageous socioeconomic circumstances. A study using Finnish and Norwegian national patient register data, however, found that direct effects of socioeconomic position on mortality among myocardial infarction patients were larger, and the socioeconomic gradient in the use of coronary revascularisation had only a minor contribution to mortality among acute myocardial infarction patients [28]. Regarding medication, better adherence to cardiovascular medication among patients in advantageous socioeconomic circumstances might widen the socioeconomic differences. Previous studies have reported that, especially, adherence to statin therapy is low among men in disadvantageous socioeconomic circumstances [9,11]. Studies about the initiation of cardiovascular medication are rare but in line with us reporting medication to be more common among people in lower socioeconomic circumstances [12,15,16] and, thus, not supporting the assumption that beginning new medication adds socioeconomic differences. Studies examining individuals with diabetes have found opposite results [13,14]. Stricter treatment guidelines for patients with diabetes might partly explain the opposite results. Hypertension [31] and, although the results have been less clear, also unfavourable blood lipid levels [32] are more common among people in disadvantageous socioeconomic circumstances, and the results were in accordance with this. Our results thus suggest that beginning new medication might reduce the socioeconomic differences in cardiovascular diseases, although poor adherence among people in disadvantaged socioeconomic circumstances might oppose this effect.

The present study included a broad framework of socioeconomic circumstances, whereas previous studies on the initiation of cardiovascular medication focused on conventional measures of socioeconomic circumstances—namely, education, occupational class, and income. In the present study, conventional measures of socioeconomic circumstances were all associated with the initiation of cardiovascular medication, none of them being paramount. Education, occupational, and household incomes are, however, not interchangeable [20], and their contributions to the initiation of cardiovascular medication might follow diverse processes [3]. Education could have increased health literacy and favourable health behaviours, whereas household income might have influenced the access to care or the possibility to afford the medication.

Previous studies have reported that childhood socioeconomic circumstances are associated with myocardial infarction [33] and cardiovascular mortality [34], although adult socioeconomic circumstances played a far more significant role. The present study supported the previous findings that childhood socioeconomic circumstances contribute to cardiovascular health. Low parental education increased the risk of lipid medication among women only, whereas childhood economic difficulties had a more widespread contribution. The associations were further adjusted for adult socioeconomic circumstances—namely, own education, occupational class, household income, housing tenure, and current economic difficulties (data not shown). The association between parental education and lipid medication among women was abolished when adjusting for own education. Otherwise, the associations between childhood socioeconomic circumstances and cardiovascular medication remained. Thus, the results suggested that the contribution of childhood socioeconomic circumstances is not solely transmitted through adult situations. Childhood socioeconomic circumstances might have direct effects on hypertension and an unfavourable lipid profile by childhood growth and health [3] or affect the development of other risk factors such as BMI and health behaviours indirectly [35].

Additionally, further material circumstances—namely, economic difficulties and housing tenure—were both associated with cardiovascular medication. Compared to conventional measures, their association with cardiovascular health has been less studied. Economic difficulties have been associated, for example, with myocardial infarction [36] and recurrent myocardial events independently of conventional measures such as education and income [37]. Non-homeownership has been associated, for example, with an increased risk of stroke [38] and coronary heart disease [39]. A shortage in material resources might result in stress responses and, thus, influence care-seeking and health behaviours or lead to neurohumoral responses involving the hypothalamic–pituitary–adrenocortical and sympatho–adrenomedullary axes [40].

BMI and health behaviours are associated with hypertension [41] and blood lipid levels [32] and with socioeconomic circumstances as well [42]. Thus, the models were adjusted for them and the associations somewhat attenuated. We ran the analysis also by adding BMI and health behaviours to the models one at a time (data not shown). It turned out that the body mass index had the greatest contribution. Overweightness was more common among participants in more disadvantageous socioeconomic circumstances. Adjusting for the BMI and health behaviours did not, however, abolish all the associations. The results thus suggest that an increased risk of cardiovascular medication was not merely due to overweightness and poorer health behaviours among individuals in more disadvantageous socioeconomic circumstances.

Studies on socioeconomic differences in cardiovascular disease have suggested that known risk factors such as smoking, hypertension, unfavourable lipid profile, diabetes, and obesity only partially explain the differences [3]. Further risk factors have been proposed, and marital status [43], work stress [44], and mental health problems [45] have been associated with cardiovascular diseases. In the present study, the models were adjusted for these risk factors, but they had only very small contributions to the associations.

Studies of patients with known cardiovascular disease instead of plain hypertension or unfavourable blood lipid levels have often reported no associations [9,12,15,46,47] or increasing medication with more advantageous socioeconomic circumstances [9,48]. In order to examine if known and rather severe cardiovascular disease contributed to the associations, the models were adjusted for register-based data of hospitalisation due to cardiovascular disease during the follow-up period. Hospitalisation had no contribution, suggesting that socioeconomic differences did not depend on the seriousness of the situation.

A model adjusting simultaneously for all covariates was also fitted (data not shown). The associations regarding childhood economic difficulties and lipid medication and childhood economic difficulties among men, low education, housing tenure, frequent current economic difficulties, and hypertension medication remained after all the adjustments. The fully adjusted models thus highlighted the importance of childhood socioeconomic circumstances and material circumstances to cardiovascular medication.

The strengths of this study included the large dataset, prospective study design, register-based data on cardiovascular medication and hospitalisation, a broad socioeconomic framework, and the possibility to adjust for several covariates. The major limitation was that we only had register-based data on cardiovascular disease and the medication and lacked data of measured blood pressure and blood lipid levels, which would have allowed comparing the receipt of medication according to the need recommended by the current guidelines. Additionally, the indication of the prescribed medication was not known, and concerning hypertension medication, this could lead to some error, as some agents are used to prevent migraine, for example. This kind of use is, however, far less common than the use for hypertension or cardiovascular disease.

There were exclusions from the linked 6603 study participants, most of which were because of receiving medication during the three years preceding the baseline survey. We ran the analyses, including these participants as a sensitivity analysis. The associations between socioeconomic circumstances and lipid medication were highly similar. For hypertension medication, the associations concerning parental education and childhood economic difficulties among women reached statistical significance, the association between childhood economic difficulties and medication among men was weaker, and the associations concerning the lowest income group and semi-professionals reached statistical significance.

Measures of socioeconomic circumstances were based mainly on self-reports, and the responses might have been influenced by factors such as health and concerning retrospective childhood socioeconomic circumstances by the current situation of the respondent. Socioeconomic circumstances were measured only at the baseline and might have changed during the follow-up. However, among those who remained employed, the occupational class was quite stable during the 12-year follow-up [49]. All participants were municipal sector employees, which limits the generalisability of the study. Measures of weight, height, and leisure time physical activity were also based on self-reports. This might dilute their contribution if employees exaggerated their physical activity and gave information that led to a wrongfully low BMI. This study was not able to cover all potential covariates. For example, a family history of cardiovascular disease might increase the likelihood of initiating cardiovascular medication by increasing the knowledge of these issues. Additionally, various comorbidities might increase the chance of medication by increasing the risk of cardiovascular diseases or increased medical appointments. As individuals in disadvantageous socioeconomic circumstances in general have poorer health, these issues might increase the medication among them.

## 5. Conclusions

The findings of our study, which showed that both conventional and further socioeconomic circumstances are associated with the initiation of both hypertension and lipid medication, expanded on those of previous research. Several measures of socioeconomic circumstances acting at different stages of the life course were associated with cardiovascular medication, with people in disadvantageous socioeconomic circumstances having higher risks. The results suggest that there are several processes behind the associations between socioeconomic disadvantage and the initiation of cardiovascular medication. The BMI and health behaviours explained the associations to some degree, but they, however, mainly remained after the adjustments. As the initiation of cardiovascular medication was more common among individuals in disadvantageous socioeconomic circumstances, the socioeconomic differences in cardiovascular diseases might be even larger without medication. Thus, there is good reason for monitoring cardiovascular medication for the optimal outcomes.

## Figures and Tables

**Table 1 ijerph-18-10148-t001:** Distribution of participants (*n*, %) and prevalence (%) of cardiovascular medication by socioeconomic circumstances.

	Hypertension Medication			Lipid Medication		
	*n*	%	% with Medication	*n*	%	% with Medication
Parental education						
High	1148	24	30	1336	23	21
Mid-level	1305	27	32	1530	27	20
Low	2386	49	34	2897	50	27
All	4839	100	33	5763	100	24
Childhood economic difficulties						
No	3720	82	31	4412	82	23
Yes	790	18	38	950	18	28
All	4510	100	32	5362	100	24
Education						
High	1427	29	29	1641	28	22
Mid-level	2555	53	32	3060	53	23
Low	868	18	41	1076	19	30
All	4850	100	33	5777	100	24
Occupational class						
Managers and professionals	1620	33	31	1888	33	23
Semi-professionals	963	20	31	1147	20	23
Non-manual employees	1630	33	34	1984	34	23
Manual workers	654	13	37	781	13	29
All	4867	100	33	5800	100	24
Household income						
Highest group	1176	25	31	1394	24	22
Second	1246	26	31	1456	26	24
Third	1359	28	35	1631	29	24
Lowest group	992	21	34	1210	21	25
All	4773	100	33	5691	100	24
Housing tenure						
Owner occupier	3233	67	32	3885	67	24
Renter	1612	33	35	1888	33	23
All	4844	100	33	5773	100	24
Current economic difficulties						
No	2576	53	31	3079	53	24
Occasionally	1874	39	33	2237	39	24
Frequently	408	8	39	474	8	25
All	4858	100	33	5790	100	24

**Table 2 ijerph-18-10148-t002:** The associations between socioeconomic circumstances and hypertension medication. Odds ratios and 95% confidence intervals.

	Base Model = Age, Gender	Base Model + BMI + Health Behaviours ^1^	Base Model + Marital Status + Working Conditions + Mental Health	Base Model + Hospitalisation Due to Cardiovascular Disease
Parental education				
High	1.00	1.00	1.00	1.00
Mid-level	1.14 (0.95–1.35)	1.11 (0.92–1.32)	1.15 (0.97–1.37)	1.13 (0.95–1.35)
Low	1.11 (0.95–1.30)	1.03 (0.88–1.21)	1.12 (0.96–1.31)	1.11 (0.95–1.29)
Childhood economic difficulties ^2^				
Women				
No	1.00	1.00	1.00	1.00
Yes	1.17 (0.98–1.41)	1.10 (0.91–1.33)	1.14 (0.94–1.37)	1.21 (1.00–1.45)
Men				
No	1.00	1.00	1.00	1.00
Yes	1.81 (1.29–2.55)	1.74 (1.22–2.48)	1.71 (1.21–2.43)	1.89 (1.34–2.68)
Education				
High	1.00	1.00	1.00	1.00
Mid-level	1.21 (1.05–1.40)	1.14 (0.98–1.32)	1.26 (1.09–1.47)	1.20 (1.04–1.39)
Low	1.58 (1.32–1.89)	1.41 (1.17–1.70)	1.70 (1.40–2.06)	1.58 (1.32–1.89)
Occupational class				
Managers and professionals	1.00	1.00	1.00	1.00
Semi-professionals	1.11 (0.93–1.32)	1.08 (0.90–1.29)	1.13 (0.95–1.35)	1.09 (0.91–1.30)
Non-manual employees	1.25 (1.07–1.46)	1.15 (0.98–1.35)	1.32 (1.12–1.57)	1.24 (1.06–1.44)
Manual workers	1.31 (1.08–1.59)	1.15 (0.94–1.40)	1.40 (1.13–1.74)	1.32 (1.08–1.60)
Household income				
Highest group	1.00	1.00	1.00	1.00
Second	1.04 (0.88–1.24)	0.98 (0.82–1.18)	1.05 (0.88–1.25)	1.06 (0.88–1.26)
Third	1.21 (1.02–1.43)	1.10 (0.93–1.32)	1.26 (1.05–1.51)	1.23 (1.03–1.45)
Lowest group	1.19 (0.99–1.43)	1.06 (0.87–1.28)	1.37 (1.07–1.73)	1.19 (0.99–1.43)
Housing tenure				
Owner occupier	1.00	1.00	1.00	1.00
Renter	1.35 (1.18–1.54)	1.20 (1.05–1.38)	1.37 (1.20–1.58)	1.35 (1.19–1.54)
Current economic difficulties				
No	1.00	1.00	1.00	1.00
Occasionally	1.19 (1.04–1.35)	1.08 (0.94–1.23)	1.17 (1.02–1.33)	1.18 (1.03–1.34)
Frequently	1.59 (1.28–1.98)	1.37 (1.09–1.72)	1.56 (1.24–1.96)	1.59 (1.27–1.99)

^1^ Health behaviours included alcohol problems, physical activity, and smoking ^2^ Not adjusted for gender.

**Table 3 ijerph-18-10148-t003:** The associations between socioeconomic circumstances and lipid medication. Odds ratios and 95% confidence intervals.

	Base Model = Age, Gender	Base Model + BMI + Health Behaviours ^1^	Base Model + Marital Status + Working Conditions + Mental Health	Base Model + Hospitalisation Due to Cardiovascular Disease
Parental education ^2^				
Women				
High	1.00	1.00	1.00	1.00
Mid-level	1.03 (0.82–1.28)	0.99 (0.79–1.23)	1.02 (0.82–1.28)	1.01 (0.81–1.26)
Low	1.34 (1.11–1.61)	1.25 (1.03–1.51)	1.32 (1.09–1.60)	1.31 (1.09–1.59)
Men				
High	1.00	1.00	1.00	1.00
Mid-level	0.95 (0.67–1.35)	0.92 (0.65–1.32)	0.96 (0.68–1.37)	0.99 (0.69–1.42)
Low	0.96 (0.71–1.31)	0.90 (0.66–1.23)	0.96 (0.70–1.31)	0.98 (0.71–1.34)
Childhood economic difficulties				
No	1.00	1.00	1.00	1.00
Yes	1.26 (1.08–1.49)	1.19 (1.01–1.40)	1.23 (1.04–1.45)	1.29 (1.10–1.52)
Education				
High	1.00	1.00	1.00	1.00
Mid-level	1.21 (1.04–1.40)	1.12 (0.96–1.30)	1.21 (1.04–1.42)	1.20 (1.03–1.39)
Low	1.34 (1.12–1.61)	1.17 (0.97–1.41)	1.33 (1.10–1.62)	1.33 (1.11–1.60)
Occupational class				
Managers and professionals	1.00	1.00	1.00	1.00
Semi-professionals	1.21 (1.01–1.45)	1.18 (0.98–1.41)	1.22 (1.02–1.47)	1.20 (1.00–1.44)
Non-manual employees	1.19 (1.01–1.39)	1.07 (0.91–1.26)	1.21 (1.02–1.44)	1.18 (1.00–1.38)
Manual workers	1.38 (1.13–1.67)	1.21 (0.99–1.48)	1.40 (1.13–1.73)	1.37 (1.12–1.67)
Household income				
Highest group	1.00	1.00	1.00	1.00
Second	1.16 (0.97–1.38)	1.10 (0.92–1.32)	1.14 (0.95–1.37)	1.17 (0.97–1.40)
Third	1.10 (0.92–1.31)	0.99 (0.83–1.19)	1.10 (0.92–1.33)	1.11 (0.93–1.32)
Lowest group	1.22 (1.01–1.46)	1.09 (0.90–1.32)	1.29 (1.01–1.64)	1.21 (1.00–1.46)
Housing tenure				
Owner occupier	1.00	1.00	1.00	1.00
Renter	1.21 (1.05–1.38)	1.07 (0.93–1.24)	1.19 (1.04–1.37)	1.21 (1.05–1.38)
Current economic difficulties				
No	1.00	1.00	1.00	1.00
Occasionally	1.15 (1.01–1.32)	1.06 (0.92–1.21)	1.13 (0.99–1.30)	1.14 (1.00–1.31)
Frequently	1.35 (1.07–1.71)	1.16 (0.91–1.47)	1.29 (1.01–1.64)	1.32 (1.04–1.67)

^1^ Health behaviours included alcohol problems, physical activity, and smoking ^2^ Not adjusted for gender.

## Data Availability

Even the anonymised dataset cannot be shared publicly, because it contains confidential medical information, and the study participants, the city of Helsinki, and the Finnish Centre for Pensions have not given their permission to data share. The data are kept on University of Helsinki computers and are available upon agreement with the Helsinki Health Study for researchers who meet the criteria for access to confidential data. Researchers interested in the data may contact the Helsinki Health Study group (email: kttl-hhs@helsinki.fi).

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
