# Peer review of "Multiple Socioeconomic Circumstances and Initiation of Cardiovascular Medication among Ageing Employees"

_ijerph, 2021, doi:10.3390/ijerph181910148_

Round 1

Reviewer 1 Report

This is a very well-done manuscript and its results are very interesting to the Public Health area. I have some questions about the study that I describe in detail below.

Introduction:

1. Lines 80 to 82 - I suggest to delete the additional objective, because it was not extensively explored in the study.

Methods:

1. Line 131: Please, delete the mention of the table 1, because in my opinion, table 1 should be firstly cited in the Results section.

2. Lines 140 to 143: Did self-reported data of weight, height, and leisure-time physical activity validate in a previously study? If yes, it should be cited in these lines. If not, it should be considered a study limitation.

Results:

1. Lines 187 to 188: The affirmative is not totally correct, because after the adjustment for BMI and health behaviours the associations of hypertension medication and occupational class, and some categories of the variables household income and current economic difficulties lost the statistical significance.

Other considerations:

1. Other considerations: I think important to fit an additional multivariate model of association between the exposures and the outcome adjusting for all potential confounding factors (age, gender, BMI, health behaviours, marital status, working conditions, mental health, hospitalization due to cardiovascular disease).

Reviewer 2 Report

To:

Editorial Board

Title: “Multiple Socioeconomic Circumstances and Initiation of Cardiovascular Medication Among Ageing Employees”

Dear Editor,

I read this manuscript and I think that:

  • Acronyms should be expressed at their first mention both in the abstract and in the main text. Please revise the entire paper.
  • Comorbidities of the patients should be also considered as they can influence results. Please discuss.
  • This is a retrospective analysis on registries. This could be considered as a limitation of the paper. Please discuss such a point in a dedicated limitation section.
  • Family history of CAD and cardiovascular risk factors should be also considered as confounder. Please include data and discuss the in the text.
  • The role of care manager should be stressed in such a context. Please discuss the paper from Ciccone MM et al. Vasc Health Risk Manag. 2010 May 6;6:297-305
